# Temporal Association of Total Serum Cholesterol and Pancreatic Cancer Incidence

**DOI:** 10.3390/nu14224938

**Published:** 2022-11-21

**Authors:** Qiao-Li Wang, Jaewon Khil, SungEun Hong, Dong Hoon Lee, Kyoung Hwa Ha, NaNa Keum, Hyeon Chang Kim, Edward L. Giovannucci

**Affiliations:** 1Department of Nutrition, Harvard T.H. Chan School of Public Health, Boston, MA 02115, USA; 2Department of Clinical Science, Intervention and Technology, Karolinska Institutet, 171 77 Stockholm, Sweden; 3Department of Medical Oncology, Dana-Farber Cancer Institute, Boston, MA 02215, USA; 4Department of Food Science and Biotechnology, Dongguk University, Seoul 04620, Republic of Korea; 5Lee Kong Chian School of Medicine, Nanyang Technological University, Singapore 308232, Singapore; 6Department of Endocrinology and Metabolism, Ajou University School of Medicine, Suwon 16499, Republic of Korea; 7Cardiovascular and Metabolic Disease Etiology Research Center, Ajou University School of Medicine, Suwon 16499, Republic of Korea; 8Department of Preventive Medicine, Yonsei University College of Medicine, Seoul 03722, Republic of Korea; 9Department of Epidemiology, Harvard T.H. Chan School of Public Health, Boston, MA 02115, USA

**Keywords:** pancreatic ductal adenocarcinoma, lipid, blood, incidence, timing

## Abstract

Previous studies have suggested a “cholesterol-lowering effect” of preclinical pancreatic cancer, suggesting lower total cholesterol as a potential diagnostic marker. Leveraging repeated measurements of total cholesterol, this study aims to examine the temporal association of total cholesterol and pancreatic cancer incidence. We conducted a nested case-control study based on a Korean National Health Insurance Service–Health Screening Cohort, including 215 pancreatic cancer cases and 645 controls matched on age and sex. Conditional logistic regression was applied to estimate the odds ratio (OR) and 95% confidence interval (CI) for the associations of pancreatic cancer incidence with total cholesterol levels across different time windows over 11 years before pancreatic cancer diagnosis (recent, mid, distant). We found that, compared to participants with total cholesterol < 200 mg/dL in the recent 3 years prior to diagnosis, those having total cholesterol ≥ 240 mg/dL showed a significantly lower pancreatic cancer incidence (OR = 0.50 (0.27–0.93)). No significant association was found in relation to total cholesterol measured in the mid and distant past. When changes in total cholesterol over the three time periods were analyzed, compared with those with total cholesterol levels consistently below 240 mg/dL over the entire period, the OR of pancreatic cancer was 0.45 (0.20–1.03) for participants with recent-onset hypercholesterolemia, 1.89 (0.95–3.75) for recent-resolved hypercholesterolemia, and 0.71 (0.30–1.66) for consistent hypercholesterolemia. In conclusion, while high total cholesterol in the recent past may indicate a lower pancreatic cancer incidence, a recent decrease in total cholesterol may suggest an elevated incidence of pancreatic cancer.

## 1. Introduction

Pancreatic cancer is the seventh most common cause of cancer death worldwide, and the vast majority of incident cases die within one year [1]. The American Cancer Society estimated that 62,210 people in the United States will be diagnosed with pancreatic cancer in 2022 and 49,830 would die from the disease [2]. With steadily increasing total deaths from pancreatic cancer, it has been predicted to be the third leading cause of cancer death in Europe by 2025 and second in the United States by 2030 [3,4]. The poor prognosis of the disease is characterized by a 5-year survival rate as low as 11%, mainly due to diagnosis at late cancer stage and resistance to therapy, indicating the utmost importance of prevention, early detection, and a better understanding of its etiology [2,5]. However, early detection of pancreatic cancer remains challenging due to the asymptomatic nature or nonspecific symptoms, e.g., abdominal pain and dyspepsia [6]. No standard screening program is available for pancreatic cancer and identified lifestyle marker and serum biomarkers have limited value in predicting the risk [7].

Previous studies have noticed a “cholesterol-lowering effect” of preclinical cancer, which suggested lower total serum cholesterol served as an indicator for undiagnosed preclinical cancers such as colorectal cancer and pancreatic cancer [8,9,10,11]. Recent laboratory work has also confirmed a marked activation of the cholesterol metabolic pathway in pancreatic cancer, resulting in an increasing transfer of cholesterol from outside to the inside of the cancer cells [12]. However, conflicting results were reported by various studies examining the association between total cholesterol level and pancreatic cancer incidence, which could be partly due to differences in the timing of cholesterol measurement across these studies [11,13,14,15,16,17,18,19]. Given the availability of data with multiple measurements of serum cholesterol level across a long time period, this study aims to clarify the relationship between serum cholesterol level and pancreatic cancer incidence by different timing of cholesterol measurements.

## 2. Methods

### 2.1. Study Design

This was a nested case-control study based on the Korean National Health Insurance Service–Health Screening Cohort (NHIS-HEALS). Detailed cohort profiles can be found elsewhere [20]. In brief, the NHIS-HEALS cohort enrolled Koreans aged 40 to 79 years in 2002, i.e., those who received the mandatory national health examination in 2002–2003. A biennial health screening was conducted for this national representative population through the year 2013. In total, 10% of all eligible participants were randomly selected and this cohort includes 514,866 Koreans [20]. Information on sociodemographic characteristics, e.g., smoking status, alcohol consumption, and physical activity, were collected through a self-reported questionnaire, and data from medical records and bio-clinical laboratory health examinations (e.g., fasting glucose and serum cholesterol level) were collected through each individual’s visits to local healthcare institutions and hospitals.

All cases of pancreatic cancer diagnosed during 2010 to 2013 were identified through medical records and reports from the Statistics Korea Registry. From the initial 514,866 participants, incident cases of pancreatic cancer of any histologic type that were diagnosed during 2010 to 2013 were identified using C25 from the International Classification of Diseases (ICD-10) code (*n* = 989). In Korea, clinical diagnosis of pancreatic cancer is made primarily through imaging technology such as computed tomography and magnetic resonance imaging [21]. As fine-needle biopsy is rarely performed for diagnosis purposes and only 10–20% of pancreatic cancer cases are resectable at diagnosis, detailed pathological data was lacking for pancreatic cancer cases [21]. Of note, the NHIS-HEALS data capture most pancreatic cancer cases in Korea, having only 0.2% difference in the case number when compared to pancreatic cancer registered in the national cancer registry of Korea [22]. From the 989 cases, we excluded cases missing serum cholesterol levels in any of the three intervals (i.e., 0 to 3 years, 4 to 7 years, and 8 to 11 years before a diagnosis of pancreatic cancer) (*n* excluded = 766) and cases without eligible control participants (*n* excluded = 8), leaving a total of 215 pancreatic cancer cases for this study. For each of the 215 cases, three control participants were selected from the initial cohort participants, based on incidence density sampling matched for age and sex on the date of cancer diagnosis of the case. Controls had no history of cancer and had measurements of serum cholesterol levels in all the three intervals. Thus, this nested case-control study included a total of 860 participants (215 cases, 645 controls).

Total serum cholesterol and blood fasting glucose levels were measured following internal and external quality control procedures [20]. Information on height, weight, and healthy lifestyle was obtained from the national health screening program database, and information on statin use was collected from the health insurance claim database. This study was approved by the International Review Board at Yonsei University College of Medicine (diary number: 4-2016-1069).

### 2.2. Statistical Analysis

To illustrate the association between the different timing of serum cholesterol level and pancreatic cancer incidence, cholesterol level was measured at three time intervals before pancreatic cancer diagnosis (i.e., index date): –11 to –8 years (distant past), –7 to –4 years (mid past), and –3 to 0 years (recent past). The mean value of cholesterol levels was used if participants had more than one measurement for each interval. Individuals’ overall cholesterol level was estimated by the cumulative average of all available cholesterol measurements. We categorized cholesterol levels into four groups according to the definition of National Institutes of Health, Ref. [23]: desirable (<200 mg/dL), borderline-high I (200–219 mg/dL), borderline-high II (220–239 mg/dL), and high (≥240 mg/dL). Association between serum cholesterol level and pancreatic cancer incidence during an overall period and at each interval was estimated by odds ratio (OR) and 95% confidence interval (CI), using conditional logistic regression stratified by matching variables (age and sex) and adjusting for cumulative average values of body mass index (BMI, <23.0, 23.0–27.4, or >27.4 kg/m^2^), smoking status (never smoker or ever smoker), alcohol (yes or no), regular physical activity (yes or no), and statin use (yes or no). Three models were applied: (1) model stratified by age and sex; (2) model 1 with further adjustment for BMI, smoking status, alcohol consumption, and regular physical activity; (3) model 2 with further adjustment for statin use, which could lower cholesterol independently of tumor status. Linear trend assessment was conducted by Wald test on a score variable created by assigning the median value to each cholesterol category. In the sensitivity analysis, to further analyze the effect of continuous increase in cholesterol level, ORs of a cumulative average of cholesterol as per 10 mg/dL increase and incidence of pancreatic cancer during the prior 3 years was calculated by multivariate conditional logistic regression. Subgroup analyses were performed to test the robustness of the association between cholesterol level and incidence of pancreatic cancer and to explore potential effect modification by predefined stratifying factors of sex, smoking, alcohol consumption, BMI, physical activity, and statin use. Unconditional logistic regression was used to adjust for the matching factors and other relevant covariates. Tests for interaction were conducted using the Wald test on the cross-products term of cholesterol level (treated as an ordinal variable) and stratifying variable (treated as a binary variable).

For analyses of temporally dynamic change of total cholesterol levels and pancreatic cancer incidence, four major patterns were defined based on two time intervals (−11 to −4 years, and −3 to 0 years): consistently low cholesterol (<240, <240 mg/dL; reference group), recent-onset hypercholesterolemia (<240, ≥240 mg/dL), recent-resolved hypercholesterolemia (≥240, <240 mg/dL), consistent hypercholesterolemia (≥240, ≥240 mg/dL). The above three models were conducted using unconditional logistic regression. The patterns were mutually adjusted in all three models.

Our previous study found that a higher fasting blood glucose change was associated with increased pancreatic cancer incidence [24]. To test the predictive value of total cholesterol on pancreatic cancer incidence independent of fasting blood glucose, we conducted a multivariable-adjusted joint analysis between the total serum cholesterol and fasting blood glucose during the prior 3 years. High cholesterol level was defined as ≥240 mg/dL and high blood glucose level as >6.1 mmol/L. All statistical tests were two-sided, with a *p*-value of <0.05 considered statistically significant. All statistical analyses were performed using SAS 9.3 (SAS Institute, Cary, NC, USA).

## 3. Results

This nested case-control study included 860 participants, of which 215 were pancreatic cancer cases and 645 were matched controls. Age-standardized cumulatively averaged characteristics of the control participants are shown in Table 1 according to the different cumulative averages of total cholesterol levels over the past 11 years. High cholesterol group participants were more likely to be women, non-smokers, irregular exercisers, and statin users.

A significantly decreased incidence of pancreatic cancer was observed with high cholesterol levels of ≥240 mg/dL in the recent past (OR = 0.50, 95% CI 0.27–0.93) but not in the overall period (−11 to 0 years) or time intervals of mid or distant past (Table 2). Further stratified analysis on recent past intervals suggested an inverse association between total serum cholesterol level of ≥240 mg/dL and lower pancreatic cancer incidence in women with OR of 0.32 (95% CI 0.13–0.80) but not in men (P_interaction_ = 0.04, Figure 1). There was no significant interaction between statin ever users and statin never users (P_interaction_ = 0.22), although in high cholesterol levels of ≥240 mg/dL, compared to those who have cholesterol level below 200 mg/dL, the OR of pancreatic cancer was 0.20 (95% CI 0.06–0.70) for those who ever used statins and 0.88 (95% CI 0.43–1.80) for those who never used statins (Figure 1). Higher cholesterol levels during the past 3 years showed a decreased incidence of developing pancreatic cancer (OR = 0.94 (95% CI 0.89–0.99) per 10 mg/dL increment) (Appendix A).

We then analyzed the association between four patterns of temporal change of cholesterol levels and pancreatic cancer incidence (Table 3). Compared with participants with cholesterol levels consistently below 240 mg/dL during the past 11 years, the OR of pancreatic cancer was 0.45 (95% CI 0.20–1.03) for those with recent-onset hypercholesterolemia (from <240 to ≥240 mg/dL), and 1.89 (95% CI 0.95–3.75) for those with recent-resolved hypercholesterolemia (from ≥240 to <240 mg/dL). Consistent hypercholesterolemia of ≥240 mg/dL was not associated with higher pancreatic cancer incidence (OR = 0.71, 95% CI 0.30–1.66). Stratified analysis indicated that such association between individual cholesterol change and pancreatic cancer incidence was not altered by participants’ characteristics of sex, smoking, alcohol consumption, BMI, physical activity, or statin use (P_interaction_ > 0.05 for all, Appendix A).

In the joint analysis, when compared to the group with low cholesterol of <240 mg/dL and low glucose of <6.1 mmol/L, participants with high cholesterol of ≥240 mg/dL but low glucose < 6.1 mmol/L showed suggestively decreased pancreatic cancer incidence of 0.63 (95% CI 0.32–1.25), whereas participants with low cholesterol but high glucose had a significantly increased incidence of 1.97 (95% CI 1.39–2.79) (Appendix A). No interaction between total cholesterol and glucose concerning pancreatic cancer incidence was detected (P_interaction_ = 0.31, Appendix A).

## 4. Discussion

In this nested case-control study with at least three repeated measurements of total serum cholesterol over 11 years in the Korean general population, we observed an inverse association between pre-diagnostic total serum cholesterol measured within 3 years before cancer diagnosis and subsequent pancreatic cancer incidence, especially for participants with a high cholesterol levels of ≥240 mg/mL. Further stratified analysis showed this inverse association was more pronounced in women than men. However, we did not observe any association between cholesterol levels measured during the time intervals of −11 to −8 years or −7 to −4 years before cancer diagnosis and incidence of pancreatic cancer. Analysis of temporal change of cholesterol showed that recent-resolved hypercholesterolemia indicated a higher incidence of pancreatic cancer. Participants with high fasting glucose but low total cholesterol in the recent past 3 years had highest incidence of pancreatic cancer. Taken together, our findings provide evidence to support total cholesterol in the recent past, but not the long-term, may be inversely associated with the incidence of pancreatic cancer.

Without repeatedly measuring serum cholesterol levels for a long-term period, most studies reported conflicting results of one-time measured cholesterol level and pancreatic cancer incidence [11,13,14,15,16,17,18,19,25,26]. For the association of total serum cholesterol level and long-term pancreatic cancer incidence, according to a previous study based on the Asia Pacific Cohort Studies Collaboration, which includes 30 cohort studies, there was no association between total cholesterol and pancreatic cancer during a median follow-up period of 6.9 years [26]. Similar null results were also reported in most other population-based studies, including studies from Finland (median follow-up of 10.2 years), Korea (mean follow-up time of 12.7 years), Lithuania (mean follow-up time of 19.3 years), and the US (excluding first 3 years’ follow-up) [14,16,17,18,19]. However, in the cohort study from the Metabolic Syndrome and Cancer Project (mean follow-up of 11.7 years), which consisted of seven cohorts from European countries, there was a significant inverse association (hazard ratio of 0.52 with 95% CI of 0.33–0.81) between total cholesterol and pancreatic cancer by comparing the highest to the lowest quintile total cholesterol level. [24] The difference of that study from other studies (including the current study) may be due to their extreme cut-offs for total serum cholesterol levels (i.e., lowest quintile of 157–187 mg/dL and highest quintile of 271–312 mg/dL). For the recent past cholesterol level and pancreatic cancer incidence, we found an inverse association between high cholesterol levels of ≥240 mg/mL within 3 years and incidence of pancreatic cancer. Further, a recent change from hypercholesterolemia to lower categories was related to higher pancreatic cancer incidence. This finding was consistent with a prior study showing an inverse association between the total cholesterol level before 0 to 12 months of diagnosis and pancreatic cancer incidence but not for over 12 months or longer [27]. In sum, our study suggested the importance of timing in the relationship between total serum cholesterol level and pancreatic cancer incidence, which may partially explain the inconsistent reported results.

The inverse association of recent past high cholesterol levels and low pancreatic cancer incidence could be mainly due to a “cholesterol-lowering effect”, as previously shown in preclinical cancers, e.g., for colorectal cancer [8,9,10,11]. In the joint analysis, among those with low glucose, high cholesterol was suggestively though not significantly associated with a lower pancreatic cancer incidence when compared with low cholesterol, which also supports the “cholesterol-lowering effect” by pancreatic cancer. Notably, this hypothesis is further supported by the biological activities of accelerated cholesterol metabolism and biosynthesis in cancer cells for building new membranes and maintaining active signaling [28]. In addition, increasing studies have demonstrated the essential role of cholesterol metabolism in various cancer cells (e.g., colon, breast, and prostate cancers), including supporting cancer progression and suppressing immune responses, which suggests cholesterol metabolism in cancer as hallmark features of tumorigenesis in general [29]. Interestingly, however, studies also found that pancreatic ductal adenocarcinoma specifically presented a markedly activated cholesterol uptake activity and overexpression of low-density lipoprotein receptors in cancer cells [12]. We found that the inverse relation did not exist prior to 3 years before cancer diagnosis, which further reinforces this possibility. Another potential mechanism could include the overexpression of interleukin-6, a pro-inflammatory cytokine, in the tumor progression of pancreatic cancer [30]. It has been proposed that inflammation induced by interleukin-6 lowers circulating levels of cholesterol [31].

We did not observe any association between high total cholesterol and pancreatic cancer incidence in the long term, nor did we find any relationship among those with consistent hypercholesterolemia, which suggests that total cholesterol may not be a causal risk factor for pancreatic cancer. A recent large cohort study reported that persistent high-density lipoprotein was independently associated with a high risk of pancreatic cancer during a median follow-up of 5.1 years when excluding participants with less than 1-year follow-up [32], indicating high-density lipoprotein being a possible risk factor for pancreatic cancer.

To the best of our knowledge, this study is the first to explore the temporal timing of total cholesterol and pancreatic cancer incidence in a long-period setting. Pre-diagnostic total cholesterol levels were repeatedly measured from blood, which facilitated an accurate risk estimate of pancreatic cancer and dynamic change patterns of total cholesterol. Nested in a large national population-based cohort, exposure information was collected prospectively and continuously, which dismissed concerns from retrospective data and underdiagnosed high cholesterol patients among the general population. We also included the most established lifestyle risk factors of pancreatic cancer in the analyses, e.g., smoking status and alcohol consumption, to minimize potential confounding effects.

Limitations of the study warrant attention. First, our inclusion criteria of having participants with at least one cholesterol measurement every 3 years led to the exclusion of participants with missing data. However, such missing data are expected to be at random regarding the ultimate diagnosis of pancreatic cancer, and we applied a random sampling of controls from a large cohort. Second, subgroup analyses were limited by statistical power in each group and category. Finer timing analysis, e.g., with a 1-year interval, was not feasible in this study. Third, due to lack of detailed clinical data, we were unable to explore possible heterogeneity in the associations by pancreatic cancer subtype. For cancer stage, 42.7% of pancreatic cases are diagnosed at localized or locally advanced stages in Korea [33], and if our finding of a positive association between recent decrease in total cholesterol and pancreatic cancer incidence holds, monitoring temporal change in total cholesterol might be a tool to capture pancreatic cancer potentially resectable at diagnosis, although the proportion is small. Furthermore, a previous study showed differential prognosis of two pancreatic head malignancies (distal cholangiocarcinoma, pancreatic ductal adenocarcinoma) despite their close anatomic proximity [34]. Future studies on the temporal associations between total cholesterol levels and pancreatic cancer subtypes defined by tumor location and histology are warranted to identify pancreatic cancer cases that are more effectively captured by time trend of total cholesterol levels and to strengthen the basis for individualized treatment plans. Lastly, to confirm the role of total cholesterol in pancreatic carcinogenesis, more studies on the potential interaction between statin use and total cholesterol are warranted. While a meta-analysis of 26 studies overall showed a reduced incidence of pancreatic cancer associated with statin use, in subgroup analyses restricted to publications with high quality scores, randomized clinical trials, or cohort studies, and those with long-term follow-up (>4 years), statin use was not significantly associated with pancreatic cancer incidence [35]. In our study, further adjustment of statin use had a little effect on the results, suggesting statin use per se may not be the main attributor to the decreased incidence of pancreatic cancer in the recent past found in our study and thus supports a possible “cholesterol-lowering effect” by pancreatic cancer itself.

Although being a most lethal disease with an over 90% death rate and less than 9% 5-year survival rate in the United States, no valid screening program and early diagnosis methods are established for pancreatic cancer [5,6]. The only serum biomarker routinely used in clinics to predict pancreatic cancer occurrence is carbohydrate antigen CA-199, with insufficient sensitivity and specificity [5]. This study results illustrated a potential clinical implication of a recent decrease in total cholesterol from ≥240 mg/dL to <240 mg/dL without lipid-lowing medication use might indicate an occult pancreatic cancer. Total serum cholesterol level has already been routinely assessed in primary healthcare for cardiovascular prevention programs due to its low cost and feasibility of testing. It could be possible to further integrate this biomarker into pancreatic cancer screening and use it for risk stratification, perhaps combined with other factors such as a change in glucose level. However, more studies with a larger sample size are expected to further verify its prevention and clinical application.

## 5. Conclusions

In conclusion, recent-past high cholesterol indicated a lower incidence of pancreatic cancer, especially among women, while recent-resolved hypercholesterolemia was related to a higher incidence. No significant association was found for high cholesterol in the distant past, which might suggest high cholesterol not being a potential etiological risk factor. This study provides insight into the early detection of pancreatic cancer with an easily tested serum biomarker.

## Figures and Tables

**Figure 1 nutrients-14-04938-f001:**
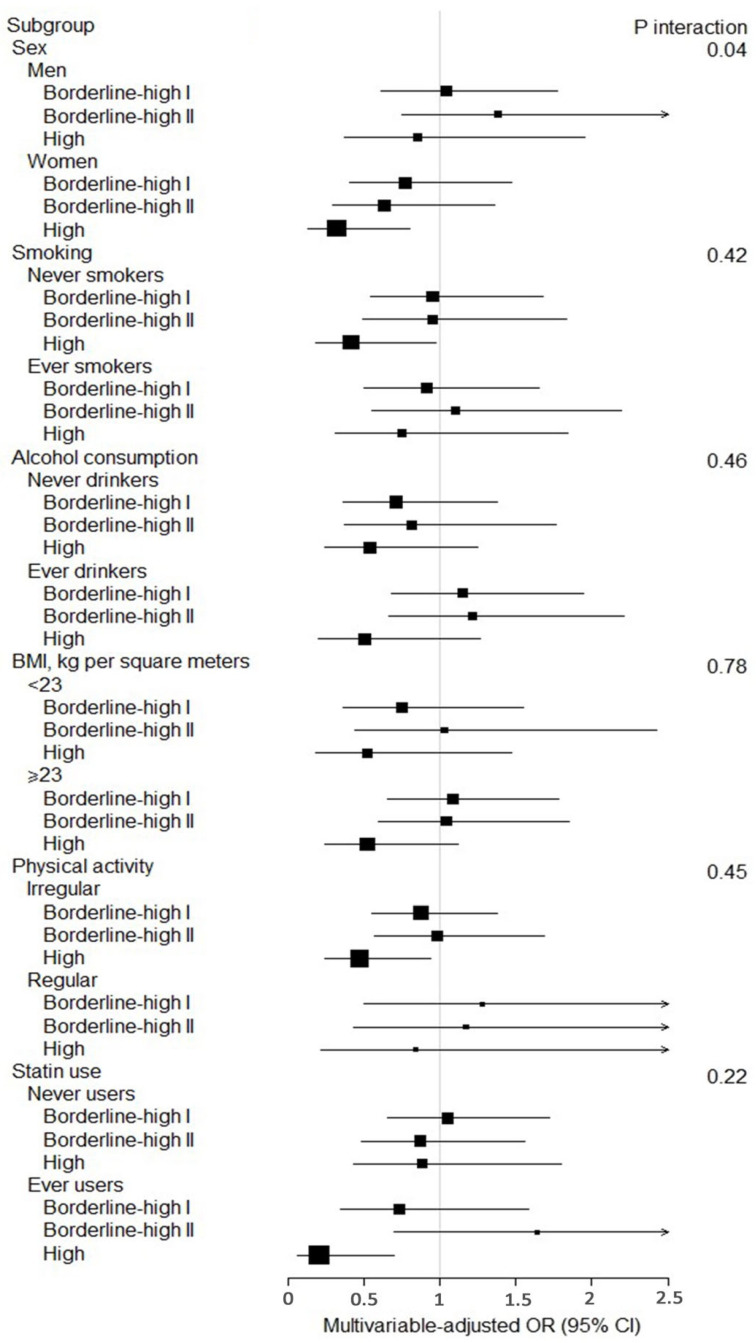
Multivariable odds ratios and 95% confidence intervals (CI) of pancreatic cancer for recent (−3 to 0 years) total cholesterol levels among subgroups, comparing borderline-high I, II, and high to desire cholesterol level. Adjusted for age (continuous, years), sex (men, women), body mass index (BMI) (<23, 23.0–27.4, ≥27.5 kg/m^2^), smoking status (never smoker, ever smoker), regular physical activity (yes, no), alcohol drinking (yes, no), and stain use (yes, no), except for the stratifying factor defining the subgroup. Abbreviation: OR = odds ratio.

**Table 1 nutrients-14-04938-t001:** Age-standardized cumulatively averaged characteristics of controls by cumulative average of total cholesterol levels over the past 11 years ^1^.

	Cumulative Total Cholesterol Levels (mg/dL)			
Characteristics ^2^	Desirable (<200)	Borderline-High I (200–219)	Borderline-High II (220–239)	High (≥240)
No of controls	340	171	92	42
Average total cholesterol levels (mg/dL)				
Overall period (−11 to 0 years)	176.4 (14.4)	209.7 (5.6)	228.6 (5.4)	261.9 (13.8)
−3 to 0 years	174.3 (21.7)	207.4 (21.0)	230.3 (23.8)	264.0 (24.7)
−7 to −4 years	174.8 (21.5)	209.0 (18.8)	226.2 (18.2)	255.1 (18.2)
−11 to −8 years	182.4 (24.9)	211.3 (22.0)	230.8 (31.7)	263.7 (29.3)
Age at the index date (years)	57.3 (9.5)	57.6 (9.1)	56.7 (9.2)	58.7 (9.2)
Women (%)	34.5	41.1	45.7	68.0
Body mass index (kg/m^2^)	23.9 (2.8)	24.3 (2.8)	24.3 (2.8)	24.8 (2.5)
Ever smokers (%)	48.8	52.0	46.5	30.5
Regular physical activity (%)	19.7	18.7	19.0	16.7
Alcohol consumption (g/day)	60.2	58.4	52.0	52.9
Statin use (%)	21.7	32.5	42.6	44.3

^1^ Characteristics were calculated as cumulative average values during the whole period. ^2^ Values are mean (standard deviation (SD)) or percentages and all, except age, are standardized to the age distribution of the control at the sampling point.

**Table 2 nutrients-14-04938-t002:** Odds ratios and 95% confidence intervals of pancreatic cancer by total cholesterol levels at different time intervals.

Time Interval of Total Cholesterol Measurement	Total Cholesterol Levels (mg/dL)	No. of Cases/Controls	Odds Ratio (95% CI) For Pancreatic Cancer		
			Age and Sex Stratified	Multivariable ^1^	Multivariable ^1^ + Statin Use
Overall period (−11 to 0 years)	Desirable (<200)	119/340	1.00 (reference)	1.00 (reference)	1.00 (reference)
	Borderline-high I (200–219)	54/171	0.90 (0.62,1.31)	0.89 (0.62, 1.30)	0.87 (0.60, 1.26)
	Borderline-high II (220–239)	27/92	0.84 (0.52,1.35)	0.83 (0.52, 1.34)	0.79 (0.49, 1.27)
	High (≥240)	15/42	1.02 (0.55,1.90)	1.01 (0.54, 1.89)	0.95 (0.50, 1.78)
	*p* for trend		0.62	0.58	0.41
Recent past (−3 to 0 years)	Desirable (<200)	131/367	1.00 (reference)	1.00 (reference)	1.00 (reference)
	Borderline-high I (200–219)	41/124	0.92 (0.62, 1.38)	0.93 (0.62, 1.39)	0.94 (0.62, 1.40)
	Borderline-high II (220–239)	28/81	1.01 (0.63, 1.61)	1.00 (0.62, 1.60)	1.02 (0.64, 1.64)
	High (≥240)	14/73	0.53 (0.29, 0.98)	0.52 (0.28, 0.95)	0.50 (0.27, 0.93)
	*p* for trend		0.12	0.1	0.09
Mid past (−7 to −4 years)	Desirable (<200)	119/359	1.00 (reference)	1.00 (reference)	1.00 (reference)
	Borderline-high I (200–219)	41/139	0.89 (0.59, 1.34)	0.89 (0.59, 1.33)	0.89 (0.59, 1.33)
	Borderline-high II (220–239)	30/81	1.12 (0.70, 1.78)	1.13 (0.71, 1.80)	1.12 (0.70, 1.80)
	High (≥240)	25/66	1.14 (0.69, 1.90)	1.12 (0.68, 1.87)	1.11 (0.66, 1.86)
	*p* for trend		0.66	0.69	0.72
Distant past (−11 to −8 years)	Desirable (<200)	107/325	1.00 (reference)	1.00 (reference)	1.00 (reference)
	Borderline-high I (200–219)	43/134	0.97 (0.65, 1.47)	0.95 (0.63, 1.43)	0.94 (0.63, 1.43)
	Borderline-high II (220–239)	30/101	0.90 (0.56, 1.43)	0.87 (0.54, 1.39)	0.87 (0.54, 1.38)
	High (≥240)	35/85	1.26 (0.80, 1.99)	1.21 (0.77, 1.92)	1.21 (0.76, 1.92)
	*p* for trend		0.55	0.7	0.71

^1^ Multivariable analyses were stratified by age (continuous, years) and sex (men, women); adjusted for smoking (never smoker, ever smoker), alcohol consumption (yes, no), body mass index (BMI) (<23.0, 23.0–27.4, ≥27.5 kg/m^2^), and regular physical activity (yes, no). Abbreviation: No. = number; Cl = confidence interval.

**Table 3 nutrients-14-04938-t003:** Odds ratios and 95% confidence intervals of pancreatic cancer by patterns of total cholesterol levels at different time windows.

Patterns ^1^ of Total Cholesterol Levels across Time Intervals (−11 to −4 Years, −3 to 0 Years)		No of Cases/Control	Age and Sex Stratified	Multivariable ^2^	Multivariable ^2^ + Statin Use
Consistently low cholesterol	(L, L)	185/550	1 (reference)	1 (reference)	1 (reference)
Recent-onset hypercholesterolemia	(L, H)	7/45	0.46 (0.20, 1.05)	0.47 (0.21, 1.06)	0.45 (0.20, 1.03)
Recent-resolved hypercholesterolemia	(H, L)	16/22	2.11 (1.09, 4.08)	2.07 (1.07, 4.02)	1.89 (0.95, 3.75)
Consistent hypercholesterolemia	(H, H)	7/28	0.74 (0.32, 1.73)	0.73 (0.31, 1.71)	0.71 (0.30, 1.66)

Abbreviation: L = lower total serum cholesterol levels (<240 mg/dL), H = higher total serum cholesterol levels (≥240 mg/dL). ^1^ Patterns of total cholesterol levels (consistently low, recent-onset hypercholesterolemia, recent-resolved hypercholesterolemia, consistent hypercholesterolemia) were mutually adjusted for in all the models. ^2^ Multivariable analyses were stratified by age (continuous, years) and sex (men, women); adjusted for smoking (never smoker, ever smoker), alcohol consumption (yes, no), BMI (<23., 23.0–27.4, ≥27.5 kg/m^2^), and regular physical activity (yes, no).

## Data Availability

Restrictions apply to the availability of these data. Data was obtained from NHIS database and are available NHIS with the permission of NHIS database.

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
