# Peer review of "Temporal Association of Total Serum Cholesterol and Pancreatic Cancer Incidence"

_nutrients, 2022, doi:10.3390/nu14224938_

Round 1

Reviewer 1 Report

The authors present a retrospective observational study of a sample of 215 patients with pancreatic neoplasia with the aim of studying the relationship between total cholesterol value and pancreatic cancer incidence. 

The introduction is well written and lays out the proper epidemiological and prognostic basis of pancreatic cancer. It also describes previous works describing the relationship between low cholesterol value and pancreatic cancer. 

Statistical methods

The authors divide the population based on total cholesterol value into 4 groups and stratify the population for some relevant characteristics (sex, age, bmi, smoking, alcohol) that could result as potential biases. The use of logistic regression with the odd ratio is correct because it is the best method to assess the risk of occurrence of an event over time. In addition, the authors perform subgroup analyses to study effects and potential biases. Congratulations on a well-structured statistical methodology. 

The results are well described and the tables are appropriate and correct.

Discussion

Well written with good contextualization of the topic.Good analysis of own results. However, some important concepts about pancreatic cancer are missing and should be integrated. 

Conclusions.

Adequate

Some considerations:

The authors discuss adenocarcinoma of the pancreas; however, how the diagnosis is made is not described. For example, do we talk about a population of operated patients? Then diagnosis made on definitive histologic examination? or diagnosis made on tc in locally advanced or metastatic patients who will never be referred for surgery and then histologic examination? This should necessarily be supplemented as it is very relevant to know the stage of the neoplasms, because the point of the study is also the lack of screening in pancreatic cancer, but if in this case the diagnoses are late anyway and the tumors metastatic, the clinical impact of this correlation is definitely less. Another key aspect is that to date the differential diagnosis between pancreatic head adenocarcinoma and distal cholangiocarcinoma is almost never completely obtainable by radiologic imaging or echoendoscopic biopsy but exclusively histological examination; therefore, the authors should clarify the method by which they obtained the diagnosis because it would be of importance and in the absence of histology they should add and discuss this limitation. To this proposed I recommend an observational study on the differential diagnosis between pdac and dCCA: 

Muttillo EM, Ciardi A, Troiano R, Saullo P, Masselli G, Guida M, Tortora A, Sperduti I, Marinello G, Chirletti P, Caronna R. Pancreatic ductal adenocarcinoma and distal cholangiocarcinoma: a proposal of preoperative diagnostic score for differential diagnosis. World J Surg Oncol. 2021 Jan 12;19(1):10. doi: 10.1186/s12957-021-02120-w. PMID: 33430887; PMCID: PMC7802249. 

Author Response

We thank the reviewer for taking time to provide us insightful comments to improve our manuscript. We revised our manuscript to address each of the comments and detailed response are provided per comment.

We agree with the comments. We elaborated on the case diagnosis/identification and discussed potential heterogeneity in the associations by tumor subtypes.

In brief, clinical diagnosis of pancreatic cancer is made primarily based on imaging technology and biopsy is rarely made rarely made for diagnostic purpose. As only 10-20% of pancreatic cancer patients are resectable at diagnosis, we lack detailed clinical data for cancer cases. Therefore, we could not assess the heterogeneity in the association by tumor subtypes as defined by stage, anatomic location, or histology. This is acknowledged as limitations in the discussion section.

The aforementioned answers are reflected on the revision as below:

Method (Page 5, Frist Paragraph)

  • In brief, the NHIS-HEALS cohort enrolled Koreans aged 40 to 79 years in 2002, those who received the mandatory national health examination in 2002-2003. A biennial health screening was conducted for this nationally representative population through the year 2013. 10% of all eligible participants were randomly selected and this cohort includes 514,866 Koreans [21].

Method (Page 5, Second Paragraph)

  • From the initial 514,866 participants, incident cases of pancreatic cancer of any histologic type that were diagnosed during 2010 to 2013 were identified using C25 from the International Classification of Diseases (ICD-10) code (N=989). In Korea, clinical diagnosis of pancreatic cancer is made primarily through imaging technology such as computed tomography and magnetic resonance imaging [22]. As fine needle biopsy is rarely performed for diagnosis purpose and only 10-20% of pancreatic cancer cases are resectable at diagnosis, detailed pathological data was lack for pancreatic cancer cases [22]. Of note, the NHIS-HEALS data capture most pancreatic cancer cases in Korea, having only 0.2% difference in the case number when compared to pancreatic cancer registered in the national cancer registry of Korea [23]. From the 989 cases, we excluded cases missing serum cholesterol levels in any of the three intervals (i.e., 0 to 3 years, 4 to 7 years, and 8 to 11 years before a diagnosis of pancreatic cancer) (N excluded=766) and cases without eligible control participants (N excluded=8), leaving a total of 215 pancreatic cancer cases for this study. For each of the 215 cases, three control participants were selected from the initial cohort participants, based on incidence density sampling matched on age and sex on the date of cancer diagnosis of the case. Controls had no history of cancer and had measurements of serum cholesterol levels in all the three intervals. Thus, this nested case-control study included a total of 860 participants (215 cases, 645 controls).

Discussion (Page 12-13, Third Paragraph) 

  • Third, due to lack of detailed clinical data, we were unable to explore possible heterogeneity in the associations by pancreatic cancer subtype. For cancer stage, 42.7% of pancreatic cases are diagnosed at localized or locally advanced stages in Korea [34], and if our finding of a positive association between recent decrease in total cholesterol and pancreatic cancer incidence holds, monitoring temporal change in total cholesterol might be a tool to capture pancreatic cancer potentially resectable at diagnosis, albeit small the proportion is. Furthermore, a previous study showed differential prognosis of two pancreatic head malignancies (distal cholangiocarcinoma, pancreatic ductal adenocarcinoma) despite their close anatomic proximity [35]. Future studies on the temporal associations between total cholesterol levels and pancreatic cancer subtypes defined by tumor location and histology are warranted to identify pancreatic cancer cases that are more effectively captured by time trend of total cholesterol levels and to strengthen the basis for individualized treatment plan.

Reviewer 2 Report

Congratulations on your study.  I would like the abstract to be divided into introduction, methods, results and conclusions.

Author Response

We thank the reviewer for taking time to provide us insightful comments to improve our manuscript. We revised our manuscript to address each of the comments and detailed response are provided per comment.

The abstract policy in this journal states that 'The abstract should be a single paragraph and should follow the style of structured abstracts, but without headings' (https://www.mdpi.com/journal/nutrients/instructions). Therefore, we kept our original format without heading, while revising transition words to better distinguish each sections of the abstract. The revised abstract is as follows.

Abstract (Page 3)

  • Previous studies have suggested a “cholesterol-lowering effect” of preclinical pancreatic cancer, suggesting lower total cholesterol as a potential diagnostic Leveraging repeated measurements of total cholesterol, this study aims to examine the temporal association of total cholesterol and pancreatic cancer incidence. We conducted a nested case-control study based on Korean National Health Insurance Service-Health Screening Cohort, including 215 pancreatic cancer cases and 645 controls matched on age and sex. Conditional logistic regression was applied to estimate the odds ratio (OR) and 95% confidence interval (CI) for the associations of pancreatic cancer with total cholesterol levels across different time windows over 11 years before pancreatic cancer diagnosis (recent, mid, distant). We found that, compared to participants with total cholesterol <200mg/dl in the recent past 3 years, those having total cholesterol ≥240mg/dl showed a significantly lower pancreatic cancer incidence (OR=0.50 [0.27-0.93]). No significant association was found in relation to total cholesterol measured in the min and distant past. When changes in total cholesterol over the three time were analyzed, compared with those with total cholesterol levels consistently below 240mg/dl over the entire period, OR of pancreatic cancer was 0.45 [0.20-1.03] for participants with recent-onset hypercholesterolemia, 1.89 [0.95-3.75] for recent-resolved hypercholesterolemia, and 0.71 [0.30-1.66] for consistent hypercholesterolemia. In conclusion, while high total cholesterol in the recent past may indicate a lower pancreatic cancer incidence, a recent decrease in total cholesterol may suggest an elevated incidence of pancreatic cancer. 

Reviewer 3 Report

The present work attempts to address the question whether high (or low) cholesterol levels are a risk factor for pancreatic cancer, using a new approach by studying the temporal effect of cholesterol levels.

This work is really interesting and it is well written and well presented by the authors.

Their question in mind is quite important since such a simple mechanistic behind pancreatic tumors could prove significant for tumor prognosis.

There are some minor issues to be addressed:

In the methodology, the authors report that 514,866 Koreans consisted of their initial cohort, while "This nested case-control study included 860 participants, of which 215 were pancreatic cancer cases and 645 were matched controls". Please explain in the "Methods" section how the sample was reduced from the initial cohort to the study-cohort.

From the way I see it, the authors suggested that low cholesterol levels short before the emergence of the tumor are considered a risk factor. They also report that cholesterol metabolism is used be cancer cells for membrane synthesis, which in an increased rate can explain lower cholesterol levels. Therefore, is it possible that this could be an overall mechanism for tumorigenesis in general, and not for pancreatic cancer alone? Please comment on that.

Finally, is it possible that statins play a protective role in pancreatic cancer or exactly the opposite? Please comment on that based on your findings.

Overall, their work has merit for publication after addressing the previous issues.

Author Response

We thank the reviewer for taking time to provide us insightful comments to improve our manuscript. We revised our manuscript to address each of the comments and detailed response are provided per comment.

  1. In the methodology, the authors report that 514,866 Koreans consisted of their initial cohort, while "This nested case-control study included 860 participants, of which 215 were pancreatic cancer cases and 645 were matched controls". Please explain in the "Methods" section how the sample was reduced from the initial cohort to the study-cohort.

Response:

We further clarified the process of selecting study participants in the Method section as below:

Method (Page 5, Second Paragraph)

  • From the initial 514,866 participants, incident cases of pancreatic cancer of any histologic type that were diagnosed during 2010 to 2013 were identified using C25 from the International Classification of Diseases (ICD-10) code (N=989). In Korea, clinical diagnosis of pancreatic cancer is made primarily through imaging technology such as computed tomography and magnetic resonance imaging [22]. As fine needle biopsy is rarely performed for diagnosis purpose and only 10-20% of pancreatic cancer cases are resectable at diagnosis, detailed pathological data was lack for pancreatic cancer cases [22]. Of note, the NHIS-HEALS data capture most pancreatic cancer cases in Korea, having only 0.2% difference in the case number when compared to pancreatic cancer registered in the national cancer registry of Korea [23]. From the 989 cases, we excluded cases missing serum cholesterol levels in any of the three intervals (i.e., 0 to 3 years, 4 to 7 years, and 8 to 11 years before a diagnosis of pancreatic cancer) (N excluded=766) and cases without eligible control participants (N excluded=8), leaving a total of 215 pancreatic cancer cases for this study. For each of the 215 cases, three control participants were selected from the initial cohort participants, based on incidence density sampling matched on age and sex on the date of cancer diagnosis of the case. Controls had no history of cancer and had measurements of serum cholesterol levels in all the three intervals. Thus, this nested case-control study included a total of 860 participants (215 cases, 645 controls).

  1. From the way I see it, the authors suggested that low cholesterol levels short before the emergence of the tumor are considered a risk factor. They also report that cholesterol metabolism is used be cancer cells for membrane synthesis, which in an increased rate can explain lower cholesterol levels. Therefore, is it possible that this could be an overall mechanism for tumorigenesis in general, and not for pancreatic cancer alone? Please comment on that.

Response:

Thank you for the great question. We replaced "risk" with "incident" and "risk factor" with "marker".Also, we added more comments in the discussion as shown below.

Discussion (Page 11, Second Paragraph)

  • Notably, this hypothesis is further supported by the biological activities of accelerated cholesterol metabolism and biosynthesis in cancer cells for building new membranes and maintaining active signaling [29]. In addition, increasing studies have demonstrated the essential role of cholesterol metabolism in various cancer cells (e.g., colon, breast, and prostate cancers), including supporting cancer progression and suppressing immune responses, which suggests cholesterol metabolism in cancer as hallmark features of tumorigenesis in general [30]. Interestingly, however, studies also found that pancreatic ductal adenocarcinoma specifically presented a markedly activated cholesterol uptake activity and overexpression of low-density lipoprotein receptors in cancer cells [12].

  1. Finally, is it possible that statins play a protective role in pancreatic cancer or exactly the opposite? Please comment on that based on your findings.

Response:

While the role of stain in pancreatic cancer development is an important question, in our study that examines the effect of cholesterol levels on pancreatic cancer risk, the direct effect of statin on pancreatic cancer risk is out of the scope while stain is an important modifier. Thus, we incorporated the reviewer’s point in the limitation section.

Discussion (Page 13, First Paragraph)

  • Lastly, to confirm the role of total cholesterol in pancreatic carcinogenesis, more studies on potential interaction between statin use and total cholesterol are warranted. While a meta-analysis of 26 studies overall showed a reduced incidence of pancreatic cancer associated with statin use, subgroup analyses restricted to publications with high quality score, randomized clinical trials or cohort studies, and those with long-term follow-up (>4 years), statin use was not significantly associated with pancreatic cancer incidence [36]. In our study, further adjustment of statin use had a little effect on the results, suggesting statin use per se may not the main attributor to the decreased incidence of pancreatic cancer in the recent past found in our study and thus supports a possible “cholesterol-lowering effect” by pancreatic cancer itself.

Round 2

Reviewer 1 Report

Accepted